# Ultrasonic Dyeing of Polyester Fabric with Azo Disperse Dyes Clubbed with Pyridonones and Its UV Protection Performance

**Alya M. Al-Etaibi** [1,*] **and Morsy Ahmed El-Apasery** [2]

1 Natural Science Department, College of Health Science, Public Authority for Applied Education and Training, Fayha 72853, Kuwait
2 Dyeing, Printing and Textile Auxiliaries Department, Textile Industries Research Division, National Research Centre, 33 El Buhouth St., Dokki, Cairo 12622, Egypt; elapaserym@yahoo.com
* Correspondence: alya.aletaibi@yahoo.com; Tel.: +96-599-807-246

**Abstract:** The textile sector is closely linked to environmental pollution as a result of the use of toxic chemicals and their disposal in liquid waste, which negatively affects for the environment. Moreover, textile industries, especially wet processing, consume a large amount of energy, water, and chemical auxiliaries. Therefore, there is an urgent need to find a solution that takes the problem of environmental pollution into account. Considering ultrasound as an environmentally safe alternative for dyeing polyester fabrics with the disperse dyes that we have prepared before, the comparison between the ultrasonic dyeing method and conventional dyeing at low temperatures was investigated. Dye exhaustion on polyester fabrics and fastness properties such as the washing, rubbing, light, and perspiration of all of the dyed fabrics were performed by two dyeing methods. Additionally, the ultraviolet protection factors (UPF) for dyed polyester fabrics were evaluated.

**Keywords:** disperse dyes; polyester fabrics; ultraviolet protection factor; ultrasonic dyeing

## 1. Introduction

Due to strict legislation and growing ecological problems, the textile dyeing industry is increasingly considering environmental issues. There are many challenges facing the textile industries, especially in the field of dyeing synthetic fabrics [1]. The main environmental impact of the textile dyeing industry involves high energy consumption. The dyeing of polyester fabric can be accomplished using two common methods. The first method is the high temperature and high-pressure dyeing method (120–130 °C). Due to high temperature conditions, this method requires high energy consumption. The second method is low-temperature dyeing under atmospheric conditions (below 100 °C) with the help of a carrier to improve the adsorption and to accelerate the diffusion of the dye into the fibers. Saving time and energy is of a great importance to textile industries. Therefore, when using practical techniques that allow the use of less energy, it is of great and urgent importance. The use of ultrasonic waves in dyeing polyester fabrics at low temperatures can be considered as a vital method of energy conservation and at the same time, an environmentally safer methodology. The long exposure to ultraviolet rays of sunlight is the main cause of sensitive skin and skin cancer. As such, the best option is to find well-designed clothes made of textiles that block UV rays. One of the characteristics that a fabric should possess is to block the Sun's rays by having a UV absorbing capability that absorbs ultraviolet rays or to be dyed fabrics where dyed fabrics provide better protection from the Sun's rays than bleached fabrics. Therefore, it can be said that the deeper the color of the fabric, the better protection from the Sun's rays that the fabric has. The use of ultrasound waves in dyeing was a benign and environmentally sustainable technique to obtain fabrics that had the ability to protect from the Sun's rays and to reduce potential skin diseases. Ultrasonic energy appears to be conceivably useful to the utilization of water-insoluble dyes with the hydrophobic fabrics [2–4]. Ultrasonic waves were utilized

in coloring polyester fabrics in contrast with traditional dyeing methods. These are high frequency longitudinal waves that disseminate into the fabrics; thus, it was expected that ultrasonic energy affects the coloring rate [5–7]. In fluid, ultrasound waves cause the cavitation or creation of microscopic bubbles. The synchronous arrangement and flattening of micro bubbles, known as cavitation, brings about an expansion in nearby high temperature, high pressing factor, and shock wave event in the color bath at the infinitesimal level. This prompted heat is large enough and sufficient for wet preparing and hence diminishes the requirement for outside warming together with improved item quality [8–18]. Additionally, ultrasound provides color penetration and dispersion that are not provided by the traditional methods; hence, it allows the fluid dissolvability of dye particles to build and cause uniform scattering of the color totals, diminishing the mean particle size of the used dyes. Ultrasonic energy helps to dye polyester fabrics as an alternate technique to the conventional dyeing of polyester fabrics, saving energy through diminishing the dyeing times and temperatures and the use of materials with expanded dyeing productivity. Ultrasonic helps the dyeing process of polyester fabrics because of the low dissolvability of the disperse dyes in the water; a lot of dispersants are expected to disperse dyes in the dyeing process, which needs numerous different synthetic substances, for example, salt and surfactants. In this study, we investigate the impact of ultrasound on the coloring behavior of polyester fabrics in order to upgrade the dye uptake, dye exhaustion, and dyeing rate for some disperse dyes.

## 2. Materials and Methods

### 2.1. Ultrasonic Dyeing

Samples of 100% polyester texture were placed into a container containing 2% disperse dye 1 or 2, the dispersing agent Matexil DA-N, and the carrier Tanavol EP 2007 at coloring temperature 80 °C and a coloring time 1 h and using a sonic force level 500 W with a wavelength 132 kHz (thermostated CREST bench top 575 HT ultrasonic,).A few of dimethylformamide was added to the dissolved dye, which was assorted with Matexil DA-N, and water was added at a liquor ratio of 1:50 at the same time that the pH was adjusted to 4.5. After the dyeing process, the colored samples were subjected to a reduction clearing process for 15 min at 60 °C.

### 2.2. Conventional Dyeing

Samples of 100% polyester texture were placed into a container containing 2% either disperse dye 1 or 2, the dispersing agent Matexil DA-N, and the carrier Tanavol EP 2007 at different conventional dyeing temperatures of either 80 °C or 100 °C for 1 h. The dyeing method was continued as described above.

### 2.3. Dyebath Exhaustions

The following equation was used to calculate the % dyebath exhaustions:

$$E\% = \frac{C1 - C2}{C1} \times 100 \tag{1}$$

where: C1 and C2 are the concentration of the dye before and after dyeing, respectively.

### 2.4. The Total Color Difference

The total color difference ΔE between the sample and the standard was calculated with the following equation:

$$E = [(\Delta L^2 + \Delta a + b^2)]^{\frac{1}{2}} \tag{2}$$

(*L*) represents the white–black axis, (*a*) represents the red–green axis, and finally, (*b*) represents the yellow–blue axis [1].

*2.5. Color Measurement*

The strength of shadings, communicated as K/S esteems, was assessed using theKubelka–Munk equation through deciding the light reflectance procedure performed on a Ultra-Scan PRO D65 UV/VIS spectrophotometer (CEM, Charlotte, NC, USA) [18].

$$K/S = \frac{(1-R)^2}{2R} - \frac{(1-R_0)^2}{2R_0} \tag{3}$$

*2.6. UPF Measurement*

It is significant that the ultraviolet protection factor is the capacity of colored polyester fabric to the hind ultraviolet that was directed in the ultraviolet visible spectrophotometer 3101.

**3. Results and Discussion**

It is well known that in high temperature dyeing, polyester fabrics need higher temperatures of over 125 °C to produce a good color penetration in the structure of the fabrics. This investigation deals with coloring of polyester fabrics by using ultrasonic energy at 80 °C.

To accomplish comparison progression for ultrasound and conventional dyeing, polyester fabrics were dyed for one hour using a 2% disperse dye concentration in presence of a carrier. The shading yield of the colored fabrics was assessed using K/S estimations since K/S values straightforwardly correspond to the concentration of the fixed color. The shading estimation data for CIE Lab was assessed based upon the $\Delta E^*$ values (total color differences). The total color differences esteems among US and conventional colored fabrics were determined using the $L^*$, $a^*$, $b^*$ values. The disperse dyes under investigation were recently synthesized by us [16] (Figure 1) and were then used to dye polyester fabrics, where orange or greenish yellow hues were obtained

**Figure 1.** Chemical structure of disperse dyes.

*3.1. Dye Uptake and Exhaustion*

The K/S color strength of the dyed polyester fabrics were expressed as dye uptakes. Many investigations [19–21] have reported that due to its sustainable and clean nature, ultrasonic energy has successfully been applied to wet textile processes, where higher color strength K/S is obtained, and this completely corresponds to what was obtained during this study. In the beginning, when we performed the dyeing process at 80 °C for both the conventional dyeing methods and the ultrasound dyeing method, the K/S value was 9.07 for the ultrasound dyeing process (see Figure 2), while the K/S value was 1.97 for the convention dying method (see Figure 2) for disperse dye 1; hence, the ratio between the two methods is 360%, which is a large percentage, so we raised the temperature of conventional dyeing method to be 100 °C.

The results in Table 1 and Figure 3 indicate that the K/S value was 9.07 in the case of dyeing using the ultrasound method at 80 °C, and it was 8.82 for dyeing using the conventional method at 100 °C, meaning that the first method using the ultrasound method was higher than the conventional method by 2.83% for the disperse dye 1. The same results were obtained for the disperse dye 2, where the K/S value was 1.27 in the case of dyeing using the ultrasound method at 80 °C, and it was 1.09 for dyeing with the conventional methods at 100 °C, meaning that the first method using ultrasound is higher than the conventional method by14.17%.From the results recorded in Table 1,it is obvious that

a positive(L*) value indicates that the disperse dye is lighter and may contain heterocycles that increase the (L*) coordinate value. This fact is maintained by the estimated value of (c*), which is positive.

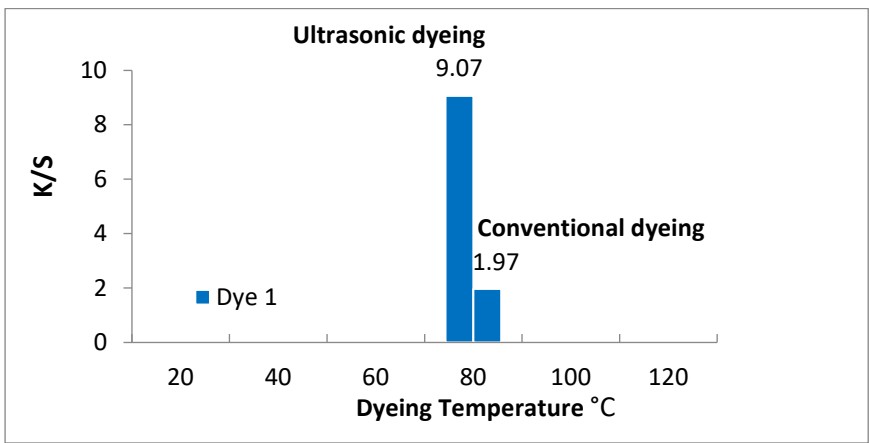

**Figure 2.** Influence of dyeing methods on dye uptakes for dye 1.

**Table 1.** Color data of the used dyes.

| Dye No | K/S | %E | L* | a* | b* | c* | h* |
|---|---|---|---|---|---|---|---|
| US dyeing at 80 °C | | | | | | | |
| 1 | 9.07 | 63.79 | 79.04 | 2.24 | 82.39 | 82.42 | 88.44 |
| 2 | 1.27 | 70.78 | 84.65 | −4.82 | 19.86 | 20.44 | 103.64 |
| Conventional dyeing at 100 °C | | | | | | | |
| 1 | 8.82 | 71.07 | 80.58 | −3.80 | 78.86 | 78.92 | 92.24 |
| 2 | 1.09 | 54.56 | 89.00 | −5.28 | 26.02 | 26.55 | 101.47 |

K is a measure of light absorption; S is a measure of light scattering; L* represents lightness; c* represents the chroma; and h* represents the hue angle.

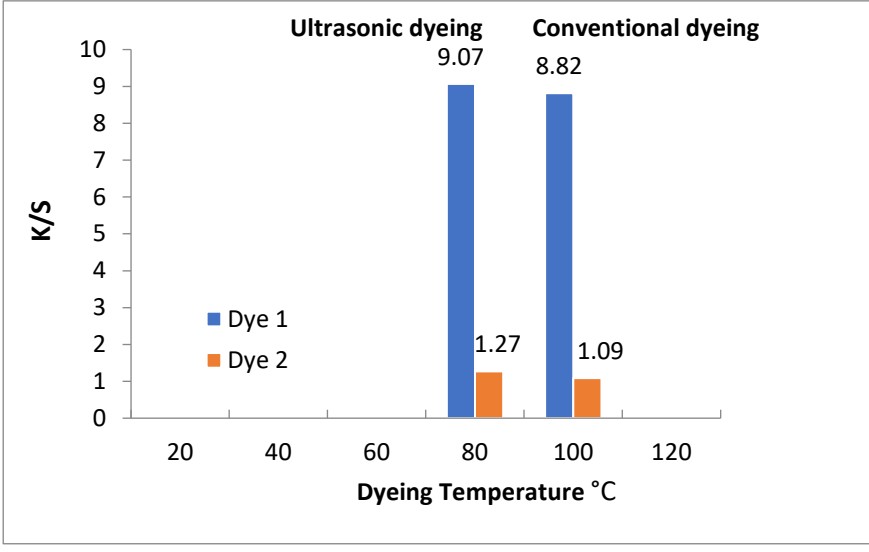

**Figure 3.** Influence of dyeing methods on dye uptakes.

Table 1 shows the percentage of dyebath exhaustion (% E) for all dyes and demonstrates very good results. The polyester fabrics dyed with disperse dyes 1 and 2 using either a conventional or ultrasonic approach exhibited satisfactory exhaustion in all circumstances and ranged from 54.56% to 71.07%.

### 3.2. ΔE (Total Color Differences)

CIE Lab and color strength K/S estimations of the colored textures were estimated for every dyeing strategy, as provided in Table 1. Total color differences ΔE* were determined, which was a solitary value which considers the contrasts between coordinates (L*, a*, b*) of the conventional and ultrasound-colored techniques as referenced by Equation (2).

The data listed in Table 1 revealed that at 2% of the color concentration for every coloring technique, the ΔE* estimations of US dyeing strategies are more prominent than ΔE* estimations for conventional dyeing techniques. The US dyed examples displayed higher ΔE* (total color differences) values of 88.61 than those obtained by conventional dyed strategies of 62.56 for disperse color 1 at the lower temperature of 80 °C within the presence of the carrier. Similar consequences for ΔE* were accomplished in US dyeing at 29.76 when compared to the conventional ones 21.30 for disperse color 2. It was additionally observed that with a decrease in lightness values and by increasing the dye uptakes, the (L*) values were consequently privileged, demonstrating more prominent shade development. For example, for the disperse dye 1, the (L*) value was 79.04 when using the ultrasound dyeing method, and then it became 80.56 when using the conventional dyeing method. Likewise, we obtained similar results for disperse dye 2, where the (L*) value was 84.65 when using the ultrasound dyeing method, and then it became 89.00 when using the conventional dyeing method.

### 3.3. Fastness Properties

The data listed in Table 2 revealed excellent perspiration fastness and very good fastness properties to light, washing, and rubbing for both the ultrasonic and conventional dyeing methods for polyester fabrics dyed with disperse dyes 1, 2.

**Table 2.** Fastness property analysis of the polyester dyed fabrics.

| Dye No | Rubbing Fastness | | Washing Fastness | | | Light Fastness | Perspiration Fastness | | | | | |
|---|---|---|---|---|---|---|---|---|---|---|---|---|
| | | | | | | | Alkaline | | | Acidic | | |
| | Wet | Dry | SC | SP | Alt | | SC | SP | Alt | SC | SP | Alt |
| US dyeing at 80 °C | | | | | | | | | | | | |
| 1 | 4–5 | 5 | 4 | 4 | 4–5 | 5–6 | 5 | 5 | 5 | 5 | 5 | 5 |
| 2 | 5 | 5 | 5 | 5 | 5 | 6 | 5 | 5 | 5 | 5 | 5 | 5 |
| Conventional dyeing at 100 °C | | | | | | | | | | | | |
| 1 | 4–5 | 5 | 4–5 | 4–5 | 4–5 | 6 | 5 | 5 | 5 | 5 | 5 | 5 |
| 2 | 5 | 5 | 5 | 5 | 5 | 6 | 5 | 5 | 5 | 5 | 5 | 5 |

Where SP = Staining on polyester, SC = Staining on cotton, Alt = Alteration.

### 3.4. Ultraviolet Protection Factor (UPF)

The recorded results in Table 3 and Figure 4 reveal that blank polyester fabric achieved a UPF value of 7.7, and the T (UV-A) esteem is 35.1; this implies that this fabric does not provide any safety against ultraviolet radiation. On the other hand, the colored polyester fabric with disperse dye 1 or 2 colored using the conventional strategy and that have 2% shade have UPF values of 66.9 and 21.3 and T (UV-A) values of 2.2 and 13.7, which infers that this texture has excellent safety against UV radiation. In Table 3, in actuality, colored polyester textures with disperse dyes 1 and 2 colored using the ultrasound technique with 2% shade have UPF esteem of 72.9 and 24.0, and the assessment of the T (UV-A) values are 2.1 and 12.4, which suggest that these fabrics have magnificent protection against the radiations of ultraviolet.

**Table 3.** UPF values of the polyester fabrics.

| Dye No | UPF | Transmittance | |
|---|---|---|---|
| | | UV-A 315–400 nm | UV-B 290–315 nm |
| Blank | 7.7 | 35.1 | 10.0 |
| US dyeing at 80 °C | | | |
| 1 | 72.9 | 2.1 | 1.1 |
| 2 | 24.0 | 12.4 | 3.0 |
| Conventional dyeing at 100 °C | | | |
| 1 | 66.9 | 2.2 | 1.3 |
| 2 | 21.3 | 13.7 | 3.4 |

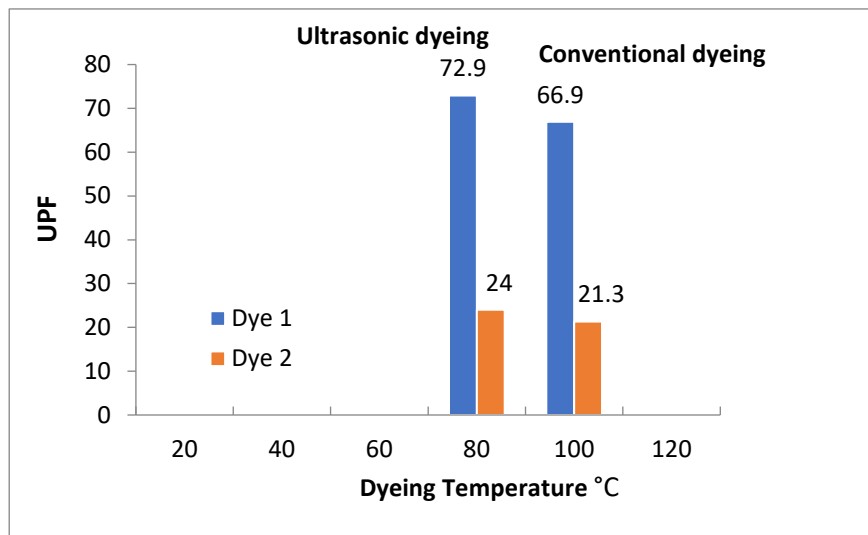

**Figure 4.** Influence of dyeing methods on UPF.

The data listed in Table 3 revealed that the ultraviolet protection factor of polyester fabrics dyed through ultrasound dyeing is better than that of those dyed with the conventional dyeing method for the used disperse dyes 1 and 2 by 8.23% and 11.25%, respectively.

## 4. Conclusions

The data indicated that the use of ultrasonic energy in the dyeing process of polyester fabrics at low temperatures leads to energy being saved. The ultrasound dyeing process is better than the conventional dyeing method, and it might be due to the fact that ultrasonic waves accelerate the diffusion of dyes into the fabrics; hence, increasing the rate of dye transfers into the fabric. The ultraviolet protection factor (UPF) of the polyester fabrics dyed with disperse dyes through the ultrasonic method is better than the UPF of the polyester fabrics dyed with disperse dyes using conventional dyeing methods. All of the disperse dyes showed very good exhaustion of the polyester fabrics, and moreover, the dyed polyester fabrics showed excellent fastness properties.

**Author Contributions:** All of the authors edited, reviewed, and wrote the manuscript. All authors have read and agreed to the published version of the manuscript.

**Funding:** This research received no external funding.

**Data Availability Statement:** Not applicable.

**Conflicts of Interest:** The authors declare no conflict of interest.

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
