# Peer review of "Ultrasonic Dyeing of Polyester Fabric with Azo Disperse Dyes Clubbed with Pyridonones and Its UV Protection Performance"

_chemistry, doi:10.3390/chemistry3030065_

Round 1
Reviewer 1 Report
The suggestions and recommendations were considered by authors and the manuscript was modified accordingly.
Author Response
Ù‹We thank the reviewer for his valuable comments
Reviewer 2 Report
In this manuscript, authors attempt to perform the sustainable disperse dyeing of polyester using the ultrasonic dyeing method. The sustainable dyeing of polyester textiles is a great challenge in the textile dyeing industry. The review has good knowledge of textile dyeing. To the best of the review’s knowledge, there are no techniques at present which can be used in the dyeing of conventional polyester fibers at low temperature below 100 ℃. So the reviewer is surprised by authors’ research ideas and data. Taking the fact into consideration that academic exchanges should be encouraged and the journal of Chemistry is an ordinary periodical, I suggest authors to make a major revision on this manuscript.
Please note the following issues as this manuscript is revised.
(1) Authors should fully explain why the low temperature can be accomplished?
(2) I think that the sustainable dyeing in the present title would mislead the readers who have no good background knowledge of textile dyeing. I do not think that the dyeing approach described in this manuscript is sustainable.
(3) Regarding the title “Sustainable dyeing of polyester with azo disperse dyes clubbed with pyridonones”: Sustainable dyeing of polyester is a big title. In addition, I do not know why authors use azo disperse dyes containing pyridonones rather than common commercial disperse dyes. If these dyes are special in dyeing properties and functions, I suggest authors to use a new title. For example:
Ultrasonic dyeing of polyester fabric with azo disperse dyes clubbed with pyridonones and its UV protection performance
Ultrasonic dyeing of polyester fabric with azo disperse dyes clubbed with pyridonones
(4) Regarding the decolorization of disperse dyes by sugarcane bagasse:
The subject of this manuscript is the dyeing of polyester with disperse dyes. I think there is no need to discuss the decolorization of residual dye solution. Indeed, in the industry of textile dyeing, the decolorization of dyes is another subject. So this manuscript should be focused on polyester dyeing. I strongly suggest authors to delete the section of dye decolorization including the relative contents in Abstract, Materials and Methods, Results and Discussion, and Conclusions.
(5) Regarding Introduction:
Authors first discuss the UV protection of textiles including dyed textiles. This is not a good writing style. The title of this manuscript is the sustainable dyeing of polyester with disperse dyes. So authors should first pay attention to the shortcomings of the present polyester dyeing.
Author Response
We thank the reviewer for his valuable comments

Reviewer 3 Report
Dear Editor,
The authors made changes and additions to their original work. The content of the paper has improved. It can be accepted for publication.
Author Response

(The authors gave the same response as above.)

Round 2
Reviewer 2 Report
The manuscript has reasonably improved. The present version can be accepted for publishing.
Author Response
Thank you
This manuscript is a resubmission of an earlier submission. The following is a list of the peer review reports and author responses from that submission.
Round 1
Reviewer 1 Report
The paper is focused on Sustainable Dyeing of Polyester with Azo Disperse Dyes Clubbed with Pyridonones.
In Materials and Methods
The author described in ultrasonic dyeing procedure that used sonic force level 500 W with 65 wavelenght at 132 kHz and pH of the dye solution was equal to 4.5. Why did the authors use these parameters? Have they been compared with those of the literature?
Authors should at least vary the ultrasonic frequency and pH of the solution, in order to have a better comparison with the results obtained in relation to conventional dyeing.
In Results and Discussion
The authors report that the shading yield of colored fabrics was assessed utilizing K/S, however these parameters are not specified. Likewise for the parameters L∗, a∗, b∗ v.
In addition, the authors report that “Total color differences ΔE* was determined, that was a solitary worth which considers the contrasts between coordinates (L∗, a∗, b∗) of the conventional and ultrasound colored techniques as referenced by equation.” What is this equation?
In according to K/S values and the total color differences ΔE∗, the author conclude that the ultrasound method is better than the conventional method. Likewise, the ultraviolet protection factor of polyester fabrics dyed with ultrasound dyeing is better than the conventional dyeing method for the used disperse dyes. However, the author must compare these data with other data of the literature.
The English should be improved.
Reviewer 2 Report
In this manuscript, authors attempt to perform the sustainable disperse dyeing of polyester using the ultrasonic dyeing method. The ultrasonic dyeing method has been reported a long time ago, but it has not been applied in practical production. Although a further academic research in this regard can be accepted, this manuscript does not display enough data in the disperse dyeing of polyester as well as new ideas. Furthermore, the exhaustion of disperse dyes is low (64-71% in Table 1) under the conditions (dye concentration 2%owf, dyeing temperature 80°C), and thus the dyeing method is not sustainable. In addition, such a low temperature (80°C) for the dyeing of polyester can not be accepted in terms of research and practical production. So I think that this manuscript has no value for being published in the journal of Molecules.
Special suggestions:
English writing needs a great improvement.
Abstract Section:
Lines 12-13: “Besides, textile industries, especially wet tissue processing, consume a large amount of energy, water and chemical auxiliaries”: “wet tissue processing” should be replaced by “wet processing”.
Introduction Section:
Lines 31-32: “Therefore, the use of ultrasound in dyeing was a benign and environmentally sustainable way to obtain fabrics that had the ability to protect from the sun's rays”: I suggest authors to delete “Therefore”.
Materials and Methods Section:
The determination of ultraviolet protection factor (UPF) should be simply described in the Materials and Methods section.
Results and Discussion Section:
Line 110: “figure 1” should be corrected into “figure 2”.
Regarding the data reuse: the same data can not be simultaneously used in the figures and tables. The data in Figures 2 and 3 are used Tables 1 and 2. The data in Figure 4 are used in Tables 4 and 5.
Reviewer 3 Report
Dear Editor,
I have read the manuscript Molecules - 1308561 entitled: “Sustainable Dyeing of Polyester with Azo Disperse Dyes 2 Clubbed with Pyridonones” and I would like to address some suggestions to the authors:
Pg. 1, line 37: ultrasonic energy affect the rate - should be - ultrasonic energy affects the rate
Pg. 1, line 46: helped dyeing polyester fabrics - should be - helped to dye polyester fabrics
Pg. 2, line 47: giving saving of energy - should be - giving a of energy
Pg. 2, line 67: What is mean “liquor ratio”?
Pg. 2, line 68: subjected to reduction clearing - should be - subjected to a reduction clearing
Pg. 2, lines 61-74: What is the difference between ultrasonic dyeing and conventional dyeing methods? Liines 62,63 are similar with lines 71, 72, please verify.
Pg. 2, lines 84: dyebath exhaustions - should be - dyebath exhaustions:
Pg. 2, lines 88: dyeing, polyester fabrics needs - should be - dyeing polyester fabrics need
Pg. 3, lines 98: What is mean L∗, a∗, b∗ values?
Pg. 3, lines 110: figure 1 - should be - Figure 1. Please verify in all manuscript.
Pg. 3, lines 118: The same results was obtained - should be - The same results were obtained
Pg. 4, Tables 1 and 2: What is mean c and h?
Pg. 4, Title 3.2. where the parenthesis closed?
Pg. 4, line 142: with decrease - should be - with a decrease
Pg. 5, line 150: table 3 - should be – Table 3 Please verify in all manuscript.
Please, compare the results obtained with others in the literature for other compounds containing disperse dyes.
Too little references are put in this manuscript.
Where it is mentioned supplementary material (non-published)?